# Hypofractionated Radiotherapy for Hematologic Malignancies during the COVID-19 Pandemic and Beyond

**Febin Antony [1], Arbind Dubey [1], Pamela Skrabek [2], Lung Fung Tsang [3], Pascal Lambert [3], Bohdan Bybel [4] and Naseer Ahmed [1,5,\*]**

1. Section of Radiation Oncology, Department of Radiology, Max Rady College of Medicine, University of Manitoba, Winnipeg, MB R3T 2N2, Canada; febinanthony@gmail.com (F.A.)
2. Section of Medical Oncology/Hematology, Department of Internal Medicine, Max Rady College of Medicine, University of Manitoba, Winnipeg, MB R3A TR9, Canada; pskrabek@cancercare.mb.ca
3. Department of Epidemiology, CancerCare Manitoba, Winnipeg, MB R3E 0V9, Canada; ltsang@cancercare.mb.ca (L.F.T.)
4. Section of Nuclear Medicine, Department of Radiology, Max Rady College of Medicine, University of Manitoba, Winnipeg, MB R3T 2N2, Canada
5. CancerCare Manitoba Research Institute, Winnipeg, MB R3E 0V9, Canada
\* Correspondence: nahmed2@cancercare.mb.ca

**Abstract:** Purpose: Radiotherapy is integral in the management of hematological malignancies (HM). Standard radiotherapy dose fractionation regimens range between 20 and 50 Gy in 10–25 fractions over 2–5 weeks. This study presents the outcomes of patients with HM treated with hypofractionation radiotherapy (HFRT) during the COVID-19 pandemic. Methods: Patients (n = 36) were treated with HFRT between January 2020 and September 2022. The outcomes measured were the overall response rate (ORR), freedom from local progression (FFLP), and overall survival (OS). Results: The median follow-up was 13.2 months. Thirty-three patients (92%) had non-Hodgkin (NHL) or Hodgkin lymphoma (HL). Eighteen patients (50%) had aggressive and nine (25%) had indolent NHL. Nineteen patients (53%) presented with stage I/II and fifteen (42%) with stage III/IV disease. Twenty-five (69.4%) and eleven (30%) received consolidative and definitive RT, respectively. Twenty patients (56%) received treatment to the neck and/or thorax and nine (25%) to the abdomen or pelvis. The total dose ranged from 18 to 42.5 Gy in 6–17 fractions/2.67–5 Gy per fraction. The median dose in 2 Gy fractions for an alpha/beta ($\alpha/\beta$) ratio of 10 amounted to 39 Gy (SD ± 13.86) and 43.6 Gy (SD ± 12) for an $\alpha/\beta$ of 3. The most commonly used fractionation scheme was 39 Gy in 13 fractions. ORR was 94.4% for the entire cohort, and 100, 94.4, and 83.3% for indolent NHL, aggressive NHL, and HL patients. The two-year FFLP was 76% (95% CI: 34–93%) for the entire cohort and 100, 87 (95% CI: 56.4–96.5%), and 42% (95% CI: 1.1–84.3%) for the indolent NHL, aggressive NHL, and HL patients. Two-year OS for the entire cohort was 80% (95% CI: 59.9–90.5%) and 100, 66.1 (95% CI: 36.4–84.4%), and 100% for the indolent NHL, aggressive NHL, and HL patients. Only one patient presented with grade two pulmonary toxicity. Conclusions: HFRT in HM provides excellent local control to be validated in a larger prospective study.

**Keywords:** lymphomas; hypofractionation; radiotherapy



## 1. Introduction

Radiotherapy (RT) is integral in the therapeutic management of hematologic malignancies (HMs), either alone, as part of a combined modality, or as a consolidation following the completion of chemotherapy. HMs include Hodgkin lymphoma (HL), non-Hodgkin's lymphoma (NHL), plasma cell neoplasms, and leukemias. The most commonly used RT dose fractionation schemes described as conventional fractionation (CF), for the treatment of HM, range between 20 and 50 Gy delivered over 2–5 weeks, with excellent outcomes [1–6].

There has been a paradigm shift in radiation oncology, with rapid advancements in RT techniques, including the introduction of four-dimensional image acquisition, real-time image guidance, and intensity-modulated radiotherapy (IMRT). As such, stereotactic ablative radiotherapy (SABR) and hypofractionated RT (HFRT) delivering highly precise and biologically effective radiation doses have been employed in common malignancies, including lung and prostate cancer [7,8]. The outcome of these contemporary interventions has been impressive, with improved local tumor control, increased overall survival, and minimal toxicity [9]. Further, the reduced number of fractions and an overall shorter treatment time may be cost effective and particularly benefit patients in remote areas away from urban cancer centers [10,11]. Therefore, it is intriguing to explore and evaluate the therapeutic efficacy of HFRT in the radiotherapeutic management of HM, assuming no increased toxicity or compromise of local control.

In March 2020, the COVID-19 pandemic was declared, creating a global healthcare crisis. Human and technical resources were reclassified and redirected to mitigate the challenges of the pandemic. RT departments were forced to rethink and innovate RT delivery models to minimize the risk of COVID-19 infection in cancer patients and healthcare staff. Consequently, several professional groups and organizations suggested alterations to conventional radiation schedules delivered over several weeks. HFRT with a reduced number of fractions, a shorter overall therapy time, and a higher dose per fraction was proposed, with or without clinical evidence, in the radiotherapeutic management of several cancers [12]. The intention was to reduce transmission and the risk of infection among immunocompromised cancer patients and involved healthcare workers, diminishing the consequences of reduced human resources during the pandemic [13]. The International Lymphoma Radiation Oncology Group (ILROG) task force provided emergency guidelines with alternative hypofractionated treatment regimens in response to the COVID-19 pandemic. The proposed dose/fractionation schemes were based on pre-defined radiobiological parameters (i.e., the $\alpha/\beta$ ratio, total dose in 2 Gy fractions (EQD2), and biological equivalent dose (BED)) to maintain the clinical efficacy and toxicity at levels similar to standard dose fractionation [14]. Currently, these recommendations are included in the guidelines of the Lymphoma Disease Site Group at our institution, to be implemented under extraordinary circumstances such as a pandemic. Here, we retrospectively evaluated the clinical outcomes of HM patients treated with HFRT at our centre during the COVID-19 pandemic, and we demonstrate that this treatment regimen is as effective in local control as standard dose fractionation.

## 2. Methods

The current study is a retrospective review of clinical outcomes in patients diagnosed with HM and treated with HFRT regimens at an academic tertiary cancer centre between 2020 and 2022 during the COVID-19 pandemic. The provincial cancer registry provided a list of patients with HM who had received RT treatment during this period. This study was approved by The Research Ethics Board (REB) of the affiliated university.

### 2.1. Data Collection

Clinical data were extracted from electronic charts using the Varian Medical Oncology (VMO) application, while RT data were collected with Varian Radiation Oncology (VRO) Eclipse 15.6. Pre- and post-treatment imaging data with accompanying text reports were captured through the IMPAX picture archiving and communication system (PACS). Two hundred and thirty-five patients who had received curative or palliative RT for HM between January 2020 and September 2022 were identified. Patients who received RT with conventional fractionation (CF) or palliative intent were excluded. Thirty-six patients with a confirmed pathological diagnosis of HM and subjected to HFRT as primary or consolidative treatment post chemotherapy were included in the analysis. Total dose in 2 Gy fractions (EQD2) for an $\alpha/\beta$ ratio of 10 and 3 was calculated for each patient [15].

*2.2. Outcome Measures*

The primary clinical outcome was overall response rate (ORR), defined as the proportion of patients with complete response (CR), partial response (PR), and stable disease (SD) within the irradiation field at the 1st follow-up after completion of treatment [16,17].

Each patient's metabolic or anatomical response was assessed based on positron emission tomography–computed tomography (PET-CT), CT scan, or magnetic resonance imaging (MRI). When pre- and/or post-treatment imaging was unavailable, the response was determined clinically. For the metabolic response, CR was defined as a Deauville score of 1, 2, and 3 with or without a residual mass. PR was defined as a Deauville score of 4 or 5 with reduced uptake compared to baseline and a residual mass of any size. SD was defined as no metabolic response with a Deauville score of 4 or 5 and no significant change in F-fluorodeoxyglucose (FDG) uptake from baseline, whereas progressive disease (PD) was defined as a Deauville score of 4 or 5 with an increase in the intensity of FDG uptake from baseline [16,17]. The anatomical response as CR, PR, SD, or PD was determined by the change in tumor size between pre- and post-treatment CT scans of the involved site, as reported by the radiologist [17].

Freedom from local progression (FFLP) was calculated from the start date of RT to the date of within-field progression by PET-CT. Patients dead without in-field progression were censored at the date of death or last follow-up [18].

Overall survival (OS) was indicated by the date of diagnosis to the date of death from any cause.

*2.3. Toxicity*

Because of the retrospective nature of this study, reliable data collection for RT-related toxicity was not possible. However, an attempt was made to capture descriptive toxicity data from the clinical notes of the treating physicians; these data were graded using the Radiation Therapy Oncology Group (RTOG) toxicity criteria [19].

*2.4. Statistical Analysis*

Summary statistics (mean, median, standard deviation) were used to describe cohort and treatment characteristics. In-field progression was calculated by the cumulative incidence function (CIF) accounting for competing risk (i.e., death). OS was assessed with the Kaplan–Meier estimator (KME).

**3. Results**

The clinical characteristics of the 36 patients included in this study are presented in Table 1. Thirty-three patients (92%) had NHL or HL; the remaining subjects included patients with peripheral T-cell lymphoma (NOS), T-cell acute lymphoblastic leukemia/lymphoma, or plasmacytoma.

The mean time from diagnosis to the start of RT treatment was 6.2 months (SD $\pm$ 2.9). The total dose ranged from 18 to 42.5 Gy in 6–17 fractions with 2.67–5 Gy per fraction. The median EQD2 for an $\alpha/\beta$ ratio of 10 and 3 amounted to 39 (SD $\pm$ 13.86) and 43.60 Gy (SD $\pm$ 12), respectively. The most frequently used fractionation scheme was 39 Gy in 13 fractions (Figure 1). The mean number of days over which treatment was completed was 12.9 days (SD $\pm$ 7.3).

Among all patients, only one subject receiving treatment to the axilla, supraclavicular, mediastinum, and bilateral hilar regions experienced RTOG grade 2 lung toxicity; however, no intervention was required. There were no reports of radiation-induced toxicity (of any grade) or treatment interruptions for the remaining patients.

**Table 1.** Clinical characteristics of the 36 included patients.

| Clinical Characteristics | *n* (%) |
|---|---|
| Mean Age | |
| 60 Years (SD 19.2).Range: 22–88 years | |
| Gender | |
| Male | 17 (47) |
| Female | 19 (53) |
| Histopathological Diagnosis | |
| NHL | 27 (75) |
| Aggressive NHL | 18 (50) |
| Indolent NHL | 9 (25) |
| HL | 6 (17) |
| Other | 3 (8) |
| Clinical Stage | |
| I | 10 (28) |
| II | 9 (25) |
| III/IV | 15 (42) |
| Missing | 2 (5) |
| Involved site of RT | |
| Cervical/mediastinum/lung/axilla | 20 (56) |
| Abdomen/pelvis | 9 (25) |
| Skin/muscle/bones | 7 (19) |
| Intent of treatment | |
| Definitive * | 11 (31) |
| Consolidative | 25 (69) |
| Response assessment | |
| PET-CT | 26 (72) |
| CT/MRI/Clinical | 10 (28) |
| RT Technique | |
| VMAT | 32 (89) |
| Electrons | 3 (8) |
| 3D-CRT | 1 (3) |

* Definitive treatment indicates patients exclusively received RT and no chemotherapy. Abbreviations: HL, Hodgkin Lymphoma; MRI, magnetic imaging resonance; NHL, Non-Hodgkin Lymphoma; PET, positron emission tomography; SD, standard deviation; VMAT, volumetric modulated arc therapy; 3D-CRT, 3-dimensional conformal radiotherapy; Electrons, electrons beam RT.

The most frequently used chemotherapy regimens were ABVD (doxorubicin, bleomycin, vinblastine, and dacarbazine) in patients with HL and R-CHOP (rituximab, cyclophosphamide, doxorubicin, vincristine, and prednisone) in those with aggressive NHL; the mean number of chemo cycles received was five (SD ± 1.5). Two HL patients had also undergone autologous stem cell transplantation. One patient with acute lymphoblastic leukemia (ALL) had been subjected to an ALL-4 chemotherapy protocol. Patients treated with definitive intent RT did not receive any chemotherapy.

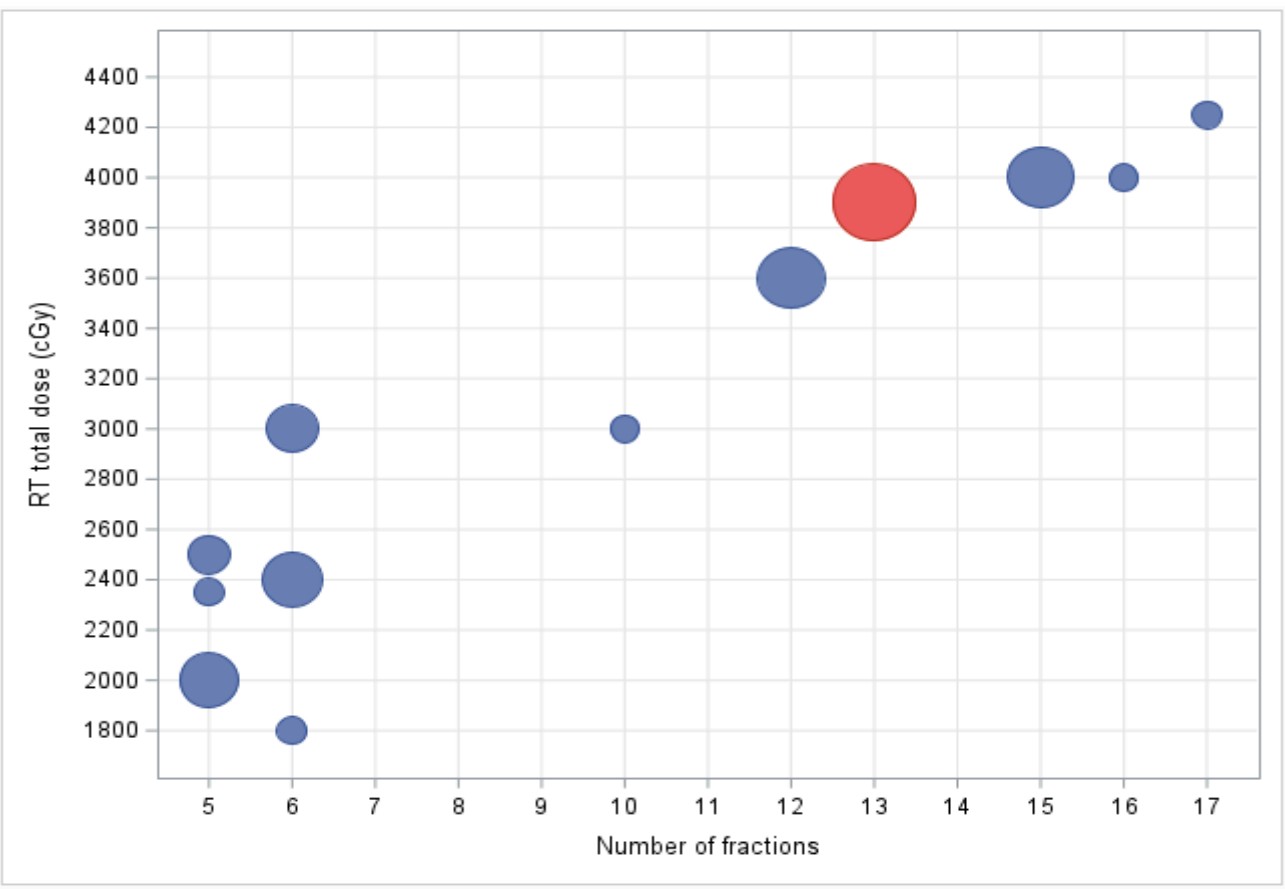

**Figure 1.** Relative frequency of dose fractionation schedules. The most commonly used fractionation scheme was 39 Gy in 13 fractions (highlighted in red). Size of each circle represents the relative number of patients treated with a particular dose fractionation scheme.

### 3.1. ORR

The mean time to first follow-up after completion of RT treatment was 2.7 months, with an interquartile ratio (IQR) of 1.28. The ORR was 94.4% for the entire cohort and 100, 94.4, and 83.3% for patients with indolent NHL, aggressive NHL, and HL, respectively. CR, PR, and SD for the entire study population amounted to 69.4, 19.4, and 5.6%, respectively.

### 3.2. FFLP

The median follow-up of the entire cohort was 13.2 months. A total of four patients (11%), two with stage IV HL and two with stage III aggressive NHL, showed in-field local progression. In addition, two patients with stage IV aggressive NHL demonstrated systemic progression without local recurrence. At two years, FFLP was 76% (95% CI: 34–93%) for the entire cohort and 100, 87 (95% CI: 56.4–96.5%), and 42% (95% CI: 1.1–84.3%) for the indolent NHL, aggressive NHL, and HL patient subgroups, respectively (Figures 2 and 3).

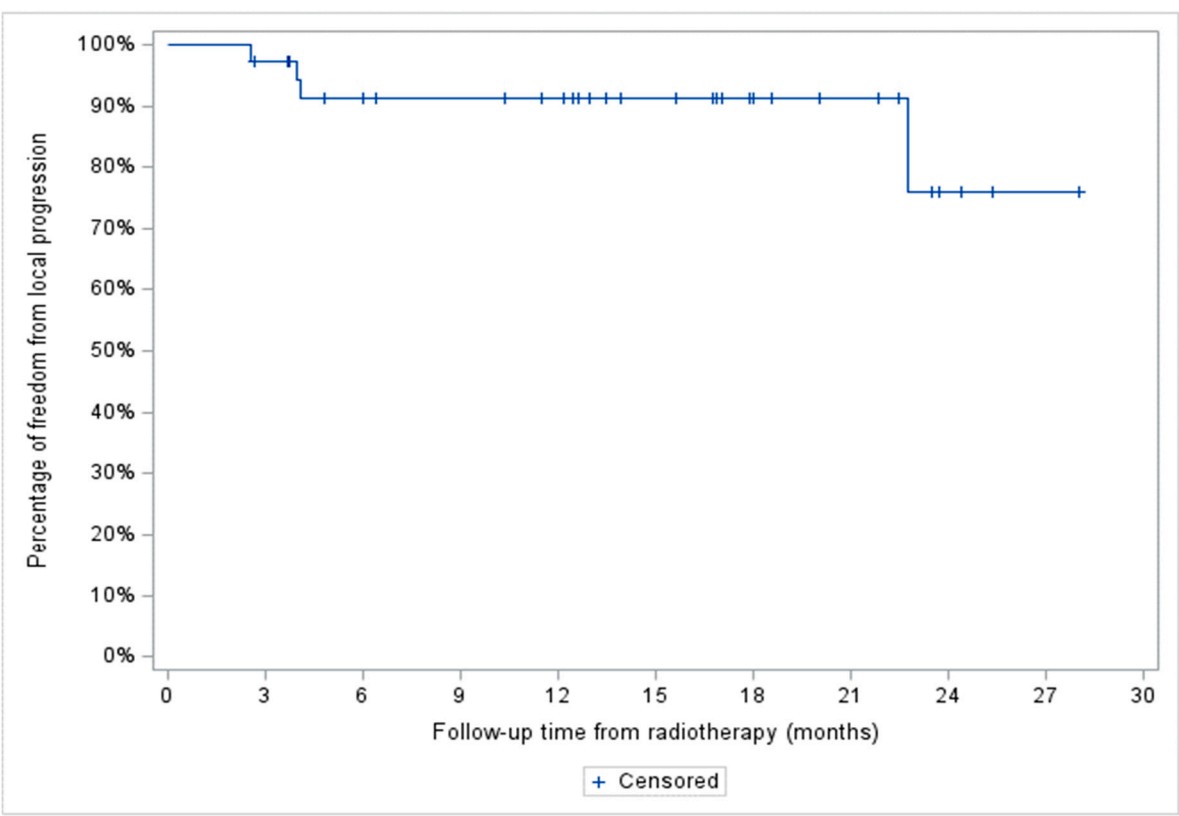

**Figure 2.** Freedom from local progression (FFLP) of the entire patient cohort (*n* = 36).

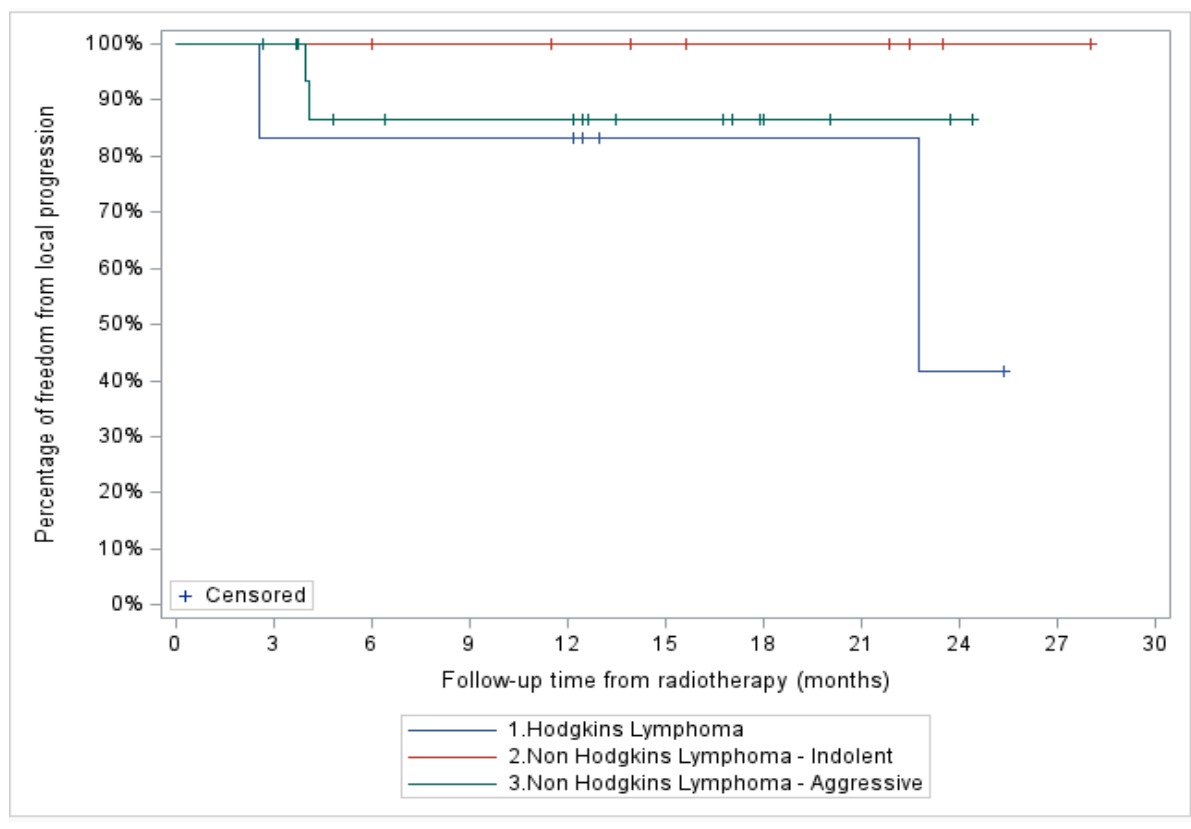

**Figure 3.** Freedom from local progression (FFLP) based on histopathological diagnosis.

### 3.3. OS

Two-year OS for the entire cohort was 80% (95% CI: 59.9–90.5%) and 100, 66.1 (95% CI: 36.4–84.4%), and 100% for the indolent NHL, aggressive NHL, and HL patient subgroups, respectively (Figures 4 and 5).

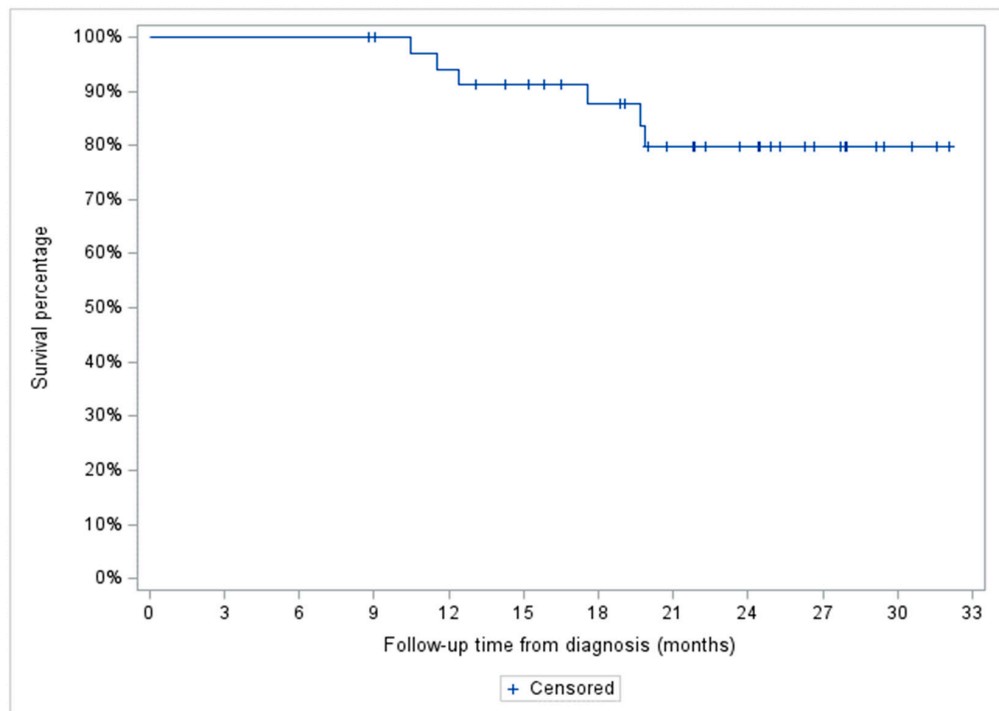

**Figure 4.** Two years overall survival of the entire patient cohort treated with hypofractionated radiotherapy (*n* = 36).

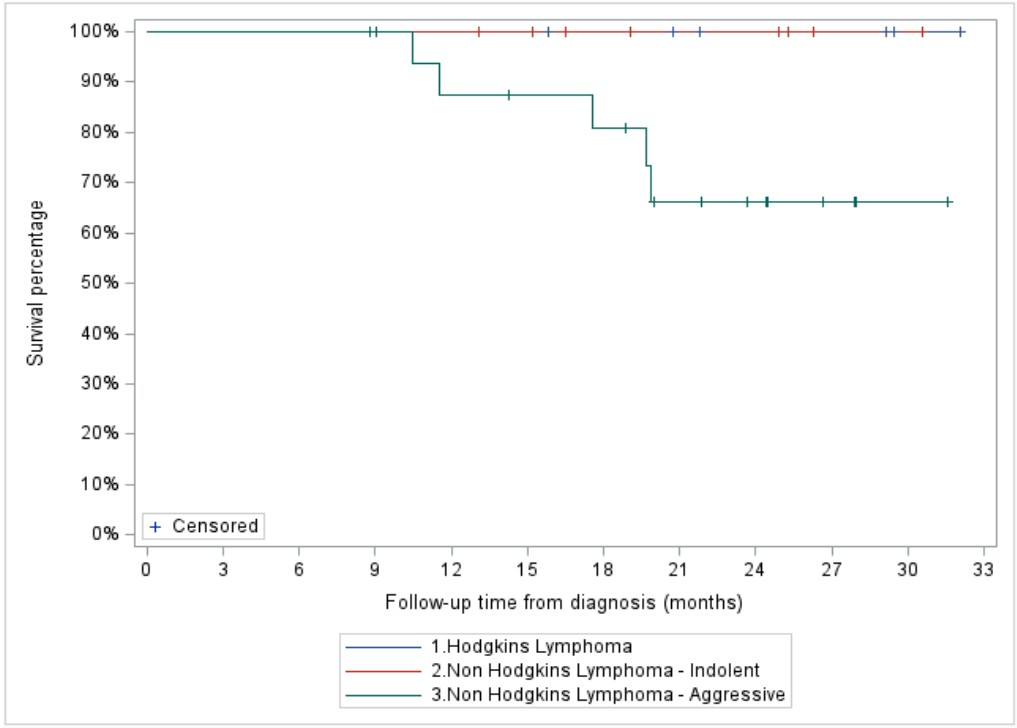

**Figure 5.** Two-year overall survival based on histopathological diagnosis.

## 4. Discussion

HMs are highly radiosensitive compared to solid malignancies, and they have a distinct radiobiological response with early interphase, premitotic, and apoptotic cell death. The possible cellular mechanisms underlying the response to RT leading to apoptosis are lipid peroxidation at the cell membrane, modulation of signal transduction, radiation-induced cross-linking of nuclear DNA, and DNA fragmentation [20]. Thus, in clinical practice, low to moderate doses involved site radiotherapy (ISRT) with CF; 20–50 Gy over 2–5 weeks) constitutes the standard radiotherapeutic management for most HM with excellent response and long-term outcomes [21]. A large randomized trial of NHL compared 24 Gy for indolent NHL and 30 Gy for aggressive NHL, with 40–45 Gy using CF. The overall response rate (ORR) was more than 90% in indolent and aggressive lymphomas, with no significant difference in the in-field progression, progression-free survival, or OS between the low- and high-dose RT [18]. Ahmed et al. demonstrated the long-term outcome of early-stage follicular lymphomas, with a CR of 100% and LC of 92%, treated with RT using CF and a median dose of 35 Gy [22]. Similarly, Hodgkin's lymphoma (HL) is extremely sensitive to RT. For early-stage disease, involved site, RT with 20–30 Gy and 2–4 cycles of Adriamycin, bleomycin, vinblastine, and dacarbazine (ABVD) are the standard of care, with excellent long-term outcomes [2,23]. For solitary plasmacytomas, RT with 40–50 Gy using CF provides a local control of 80–90% [24].

The current study is a review of patients treated with HFRT, with an EQD2 similar to the standard dose fractionation. However, the interconversion of fractionation schemes comes with uncertainties and has been challenging. The Linear Quadratic (LQ) model is the most widely used radiobiological model to calculate BED and EQD2 for different dose fractionation schedules in clinical practice, incorporating both mitotic and apoptotic cell death, and can predict the radiation effects on tumor control and normal tissue complications as a consequence of altered dose fractionation schedules [25]. The LQ model assumes two components of cell killing: one proportional to dose ($\alpha$, linear, and dose rate independent) and the other proportional to the square of the dose ($\beta$, quadratic, and dose rate dependent). Thus, the $\alpha/\beta$ ratio (ABR) incorporated into the LQ equation reflects the numerical expressions of radiosensitivity and provides a tool to determine BED and EQD2 for tumor control and normal tissue toxicity for hypo- or hyperfractionation in comparison to conventional fractionation [26–28]. The radiosensitivity of lymphomas is remarkable, with a low surviving fraction at 2 Gy (SF2Gy), a high ABR (8–10 Gy), and little to no shoulder on a cell survival curve [29–32]. During the COVID-19 pandemic, the ILROG designed a set of HFRT dose fractionation schedules to treat HM based on EQD2 with an ABR of 10 and 3 for measurable endpoints of tumor control as well as acute and late toxicity [14]. With the provided radiobiological parameters for dose conversion, HFRT is expected to achieve similar tumor control as conventionally fractionated radiotherapy (CF-RT), and any potential late toxicity for organs at risk (OAR) within the irradiation field may be mitigated by the modern conformal RT techniques. To date, there is scant literature available on the efficacy of HFRT in HM.

Wright et al. reported the outcomes of 169 patients with relapsed or refractory diffuse large B-cell lymphoma treated with salvage and palliative RT. One hundred RT courses (49%) were delivered with HFRT with a median RT dose of 30 Gy (8–60 Gy). The ORR was 60% for the entire cohort. No statistically significant differences were observed in ORR, time to local failure (TTLF), and OS between hypofractionation and conventional fractionation [33]. Takahashi et al. retrospectively reviewed the outcomes of 162 patients with aggressive NHL treated with radical, consolidative, or palliative RT. HFRT (2.4–3 Gy daily fractions; median total dose 30 Gy/10 fractions) and CF (1.8–2 Gy daily fractions; median total dose 40 Gy/20 fractions) were used in 51 and 111 patients, respectively. No differences in ORR, FFLR, and OS were observed between the two groups [34]. A very recent publication reported the effects of HFRT in gastric lymphoma. Forty-five patients with localized gastric mucosa-associated lymphoid tissue (MALT) lymphoma received 30–36 Gy in 15–18 fractions, 26–28 Gy in 13–14 fractions, or 24–25 Gy in 10 fractions.

The results revealed excellent local control and survival, with no serious adverse events, regardless of dose fractionation [35]. Jiaqi Fan et al. demonstrated the potential benefits of HFRT in a small number of patients as a bridging therapy in the context of CAR-T cell therapy in relapsed or refractory lymphoma for local control, without any significant toxicity [36].

The current retrospective study is unique in several aspects. It was conducted at an academic Centre, where patients with HM qualifying as candidates for RT were selected through a weekly multi-disciplinary conference; most RT plans go through a peer-reviewed quality assurance process. The majority of patients included in the study were diagnosed with NHL. Only those patients who received RT with an intention of local control and/or cure were considered. Though most patients received RT to cervical and/or thoracic sites, a small number of patients (17%) were exclusively treated to their abdomen, including retroperitoneum, mesentery, and spleen, without any documented toxicity. Patients who underwent consolidative RT following chemotherapy received standard chemotherapy according to institution guidelines [37].

Most patients were treated with modern techniques using VMAT. PET-CT was used in the majority of patients to assess their response to treatment. The HFRT dose fractionation used in this study is similar or close to the EQD2 for standard CF schedules and ranged between 39 and 43 Gy, with clinical outcomes similar to previously published data [2–4]. The current study, mainly comprising NHL patients, showed an ORR of 94% at approximately 3 months post-completion of HFRT treatment, demonstrating no compromise in ORR with this approach. We demonstrate excellent in-field local control with HFRT, as reflected by FFLP scores of 87–100% at two years for all NHL patients. The inferior FFLP for HL in contrast to historical data is more likely a statistical glitch, possibly because of the very small size of this group of patients in the study.

It is not feasible to compare the outcome of the patients in the current study with historical data, where patients have been mostly treated with CF, older RT techniques with 2D or 3D conformal radiotherapy, and large treatment fields. Modern highly conformal RT techniques and treatment volumes limited to the involved site are promising, with potentially reduced acute and late toxicity with HFRT without compromising local control [1,38].

*Limitations*

The authors acknowledge that the current results are based on a single-institution retrospective study involving a relatively small sample size and short follow-up; we aim to complement our current findings with results from larger patient groups and longer-term follow-ups, once data are available. Due to the small sample size, subset analysis based on histopathology, dose fractionation, disease stage, and treatment intent was impossible. Specifically, the number of patients with HL and non-lymphoma HM was extremely low for any meaningful analysis. A major limitation is the absence of structured criteria to capture and document toxicity, compounded further by the posed logistic challenges to collect toxicity data during COVID-19. Only long-term follow-up will determine the risk of late toxicity in younger patients.

## 5. Conclusions

HMs are highly radiosensitive, with excellent response and local control achieved with HFRT. The current study provides robust pilot data for a prospective multicenter trial to confirm and validate the therapeutic effectiveness and lack of toxicity of curative intent HFRT in HM.

**Author Contributions:** F.A.: formal analysis, data curation, writing—original draft; A.D.: writing—review and editing, supervision, resource; P.S.: writing—review and editing; L.F.T.: software, formal analysis, data curation; P.L.: resource; B.B.: resource; N.A.: conceptualization, methodology, formal analysis, writing—final draft, writing—review and editing, project administration. All authors have read and agreed to the published version of the manuscript.

**Funding:** This research received no external funding.

**Institutional Review Board Statement:** The study was conducted in accordance with the Declaration of Helsinki and approved by the Institutional Review Board (or Ethics Committee) of the University of Manitoba, HS25664 (H2022:287)) 29 September 2022.

**Informed Consent Statement:** Patient consent was waived as this was a retrospective chart review of the patients who have all been treated.

**Data Availability Statement:** The data presented in this study are available on request from the corresponding author.

**Acknowledgments:** Oliver Bucher from the department of Epidemiology, CancerCare Manitoba, Winnipeg, Manitoba, Canada, for his input regarding the study statistics.

**Conflicts of Interest:** The authors declare no conflicts of interest.

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
