# Peer review of "Hypofractionated Radiotherapy for Hematologic Malignancies during the COVID-19 Pandemic and Beyond"

_curroncol, doi:10.3390/curroncol31010025_

Round 1
Reviewer 1 Report
Comments and Suggestions for Authors
The authors analyzed the usefulness of hypofractionated radiotherapy (in contrast to standard dose fractionation) in the treatment of hematological malignancies, specifically Hodgkin and Non-Hodgkin lymphomas. The data are of interest. The authors address appropriately the main problems and limitations of the study which are small number of cases, retrospective study, lack of a direct comparison, short follow-up, etc. Hence this study qualifies as a pilot project.
Minor Points:
(1) Table 1: Line „Clinical Stage, Missing“: there is no number.
(2) Define "CFRT" on page 9, line 3.
Author Response
Please see the suggested changes have been applied and highlighted
1) Table 1: Line „Clinical Stage, Missing“: there is no number.
Number added
(2) Define "CFRT" on page 9, line 3.
Defined as conventionally fractionated radiotherapy ( CFRT)
Reviewer 2 Report
Comments and Suggestions for Authors
This paper is difficult to review because line numbers were not provided in the text. The topic is of interest as hypofractionation is being implemented for many sites, but this study is very small and has short follow-up that limits what can actually be determined, so this particular study is not very informative. Specific comments on the text:
- The last sentence of the abstract concludes that HFRT "provides excellent local control without any increase in toxicity." That statement cannot be supported by the study data and should be modified. In the text, the authors say "Because of the retrospective nature of this study, reliable data collection for RT-related toxicity was not possible." So there may have been toxicities that were not captured for these patients. Also, the follow-up time is short, median 13.2 months, so there may be an increase in late toxicities that hasn't occurred yet.
- Second paragraph of the Introduction: there should be references provided for the increased use of SBRT and HFRT for lung and prostate. Also, there should be references more relevant to HM; early-stage lung cancers and prostates do not seem representative of most HM RT sites. Really here should be the citations specific to HM which are currently given in the Discussion.
- Discussion: most of the first three paragraphs of the Discussion should actually be in the Introduction as it supports the reason and prior evidence around HFRT. Re-write the introduction to include this information and remove it from the Discussion.
- Table 1 has an empty column; the mean age result is given in the headings column. Also should provide the range of ages in addition to the Mean and SD. "Missing clinical stage" result is given as S instead of a number. 3D and electrons should be listed under "RT Technique" with the number of patients treated with each modality given. The paragraph after the table just lists much of the data given in the table and the portions redundant with the table should be removed.
- Figure 1: there should be a legend showing how many patients each size of circle corresponds with. Or the numbers should be printed on the circles, or the data should be given in a table rather than in this figure.
- Discussion: the discussion should really be comparing the results of this study with historical controls. The authors admit that one limitation of this study is that they did not compare their results to a patient group treated with CF, but they do not say WHY they don't do that. It would greatly strengthen the conclusions of the study that HFRT provides the same local control with no added toxicities.
Author Response
Thanks for your comments and suggestions. Please see my response below.
The last sentence of the abstract concludes that HFRT "provides excellent local control without any increase in toxicity." That statement cannot be supported by the study data and should be modified.
The sentence has been modified and highlighted in the manuscript as below
HFRT in HM provides excellent local control to be validated in a larger prospective study.
Second paragraph of the Introduction: there should be references provided for the increased use of SBRT and HFRT for lung and prostate.
Additional references have been added and highlighted. # 7 & 8.
Also, there should be references more relevant to HM; early-stage lung cancers and prostates do not seem representative of most HM RT sites. Really here should be the citations specific to HM which are currently given in the Discussion.
Please see references 1-6, including an editorial (reference#1 ) published in Red Journal about the role of radiotherapy in hematological malignancies. There is very scant or no literature about hypo-fractionated radiotherapy specifically for curative intent in hematological malignancies as described in the discussion section.
Discussion: most of the first three paragraphs of the Discussion should actually be in the Introduction as it supports the reason and prior evidence around HFRT. Re-write the introduction to include this information and remove it from the Discussion.
The primary reason for using HFRT in hematological malignancies at our center was to mitigate the COVID-19 crisis, as described in the introduction—1st three paragraphs in the discussion describe the radiobiological response of hematological malignancies. An additional paragraph (highlighted yellow) has been added in discussion, for the outcome of conventional dose fractionation in hematological malignancies.
Table 1 has an empty column; the mean age result is given in the headings column. Also should provide the range of ages in addition to the Mean and SD. "Missing clinical stage" result is given as S instead of a number. 3D and electrons should be listed under "RT Technique" with the number of patients treated with each modality given. The paragraph after the table just lists much of the data given in the table and the portions redundant with the table should be removed.
Table 1 empty column deleted. Age range has been added. Missing stage has been added. Number of patients with with Electrons and 3D have been added in table 1.
Figure 1: there should be a legend showing how many patients each size of circle corresponds with. Or the numbers should be printed on the circles, or the data should be given in a table rather than in this figure.
A sentence has been added below the figure 1. Size of each circle represents the relative number of patients treated with a particular dose fractionation scheme.
Discussion: the discussion should really be comparing the results of this study with historical controls. The authors admit that one limitation of this study is that they did not compare their results to a patient group treated with CF, but they do not say WHY they don't do that. It would greatly strengthen the conclusions of the study that HFRT provides the same local control with no added toxicities.
A paragraph has been added in discussion with references as below,
It is not feasible to compare the outcome of the patients in the current study with historical data, where patients have been mostly treated with CF, older RT techniques with 2-dimensional or 3-dimensional conformal radiotherapy and large treatment fields. Modern highly conformal RT techniques and treatment volumes limited to the involved site are promising, with potentially reduced acute and late toxicity with HFRT without compromising local control [1,41].
Reviewer 3 Report
Comments and Suggestions for Authors
In the study conducted by Antony et al., titled "Hypofractionated Radiotherapy for Hematologic Malignancies during the COVID-19 Pandemic and Beyond," the authors retrospectively assessed the clinical outcomes of 36 patients with hematological malignancies (HM) who underwent hypofractionation radiotherapy (HFRT) between January 2020 and September 2022. The study aims to establish the efficacy of HFRT in achieving comparable local control outcomes to those achieved with standard dose fractionation.
While the paper addresses an important topic, several critical points require clarification and additional information:
1. The rationale behind the selection of hematological malignancies (HM) patients for hypofractionation radiotherapy (HFRT) is not clearly delineated. The authors should provide a comprehensive explanation of the underlying reasons and considerations that led to the application of HFRT in this specific patient population.
2. The paper lacks an essential component in the form of a detailed presentation of the clinical and peripheral blood profiles of the HM patients. Including this information is crucial for a comprehensive understanding of the patients' overall health status and its potential impact on treatment outcomes.
3. The manuscript does not provide information on peripheral blood counts both before and after HFRT. Inclusion of this data is vital for assessing the treatment's impact on hematologic parameters and understanding any potential correlations between peripheral blood counts and clinical outcomes.
Addressing these concerns will significantly enhance the clarity and completeness of the manuscript, ensuring its robustness and contributing to the overall quality of the research presented.
Author Response
The paper lacks an essential component in the form of a detailed presentation of the clinical and peripheral blood profiles of the HM patients. Including this information is crucial for a comprehensive understanding of the patients' overall health status and its potential impact on treatment outcomes.
The manuscript does not provide information on peripheral blood counts both before and after HFRT. Inclusion of this data is vital for assessing the treatment's impact on hematologic parameters and understanding any potential correlations between peripheral blood counts and clinical outcomes.
Authors acknowledge the comments and suggestions.
With the given small sample size of the study, suggested analysis is not feasible. A prospective study where blood samples be collected at predetermined, discrete time intervals; effects of HFRT on peripheral blood profile and apoptotic biomarkers could be determined.
Reviewer 4 Report
Comments and Suggestions for Authors
The article is overall well written and covers an interesting topic
Nonetheless, I would suggest the following revisions
Main revisions
Although the results of this experience are promising, there are two issues I would highlight
1. Results for HL were suboptimal and worse than those reported in literature and current practice. This is likely due to the small sample, but I would add a paragraph illustrating potential explanations
2. Hypofractionation in my opinion up to date could not be considered a standard option for all patients, but only for selected cases
2018 guidelines for r/r DLBCL doi.org/10.1016/j.ijrobp.2017.12.005 propose HFRT only for patients with a limited life expectancy
ILROG emergency guidelines during the COVID-19 pandemic doi.org/10.1182/blood.2020006028 suggested the adoption of HFRT for a larger sets of patients, but highlighting to weigh the potential impact of hypofractonated schedules, also considering the generally lower α/β of OARs compared with lymphoma.
As hypofractionation has not been rigorously tested in prospective randomized trials in the curative setting, the recommendation is to use it only in the pandemic situation for curative treatment.
Nonetheless, HFRT has been increasingly used since Covid-19 spread doi.org/10.1002/hon.3151
And multiple experiences reported promising results in term of disease control and toxicity
using highly HFRT for NHL treatment with definitive intent doi.org/10.1182/blood-2023-191089
and in r/r b-cell lymphoma https://doi.org/10.1182/blood-2023-184808
also in relatively large cohorts with no difference with conventional fractionation doi:10.1016/j.adro.2022.101016
HFRT have a consolidated role as bridging treatment for CAR-T therapy doi:10.3390/cancers15102751. doi:10.3324/haematol.2023.282804. doi:10.1016/j.radonc.2023.109580.
and multiple reports suggest its safety and effectiveness in combination with immunotherapy for HL doi: 10.1016/j.clml.2021.09.005.
Therefore, I would expand this in the discussion with the provided and/or further references. In my opinion, HFRT (maintaining an EQD2 comparable with approved doses) could be clinically adopted as a standard for CAR-T bridging, palliation and unfit patients. Its standard use for fit patients with curative intent should be confirmed by randomized trials
ORR is generally considered as CR+PR. Please explain why was SD considered for ORR
Moreover, it’s not clear to me how Deauville 5 could not be considered as PD as the definition of DS5 is “markedly increased uptake or any new lesion (on response evaluation)”
Minor revisions
“of contemporary interventions”: clarify what interventions
Specify is anatomical response and metabolic response were defined according to RECIST and PERCIST criteria
Results
Specify the number of patients treated with definitive intent and the type of indolent NHL
Specify the timing of chemotherapy
Discussion
“Others retrospectively reviewed” specify the name
“modern technology” replace with “modern techniques”
Author Response
Results for HL were suboptimal and worse than those reported in literature and current practice. This is likely due to the small sample, but I would add a paragraph illustrating potential explanations.
Added and highlighted on page 10 in discussion section as below.
The inferior FFLP for HL in contrast to historical data is more likely a statistical glitch, possibly because of the very small size of this group of patients in the study.
Hypofractionation in my opinion up to date could not be considered a standard option for all patients, but only for selected cases. 2018 guidelines for r/r DLBCL doi.org/10.1016/j.ijrobp.2017.12.005 propose HFRT only for patients with a limited life expectancy
In the given reference dose fractionation is in the range of conventional fractionation.
ILROG emergency guidelines during the COVID-19 pandemic doi.org/10.1182/blood.2020006028 suggested the adoption of HFRT for a larger sets of patients, but highlighting to weigh the potential impact of hypofractonated schedules, also considering the generally lower α/β of OARs compared with lymphoma.
Please see reference 14 in our manuscript. Yahalom, J.; Dabaja, B.S.; Ricardi, U.; Ng, A.; Mikhaeel, N.G.; Vogelius, I.R.; Illidge, T.; Qi, S.; Wirth, A.; Specht, L. ILROG emergency guidelines for radiation therapy of hematological malignancies during the COVID-19 pandemic. Blood 2020, 135, 1829-1832.
As hypofractionation has not been rigorously tested in prospective randomized trials in the curative setting, the recommendation is to use it only in the pandemic situation for curative treatment.
Our study is a pilot study and provides foundation for a larger prospective clinical trial to test the efficacy and toxicity of HFRT in hematological malignancies.
Nonetheless, HFRT has been increasingly used since Covid-19 spread doi.org/10.1002/hon.3151
In the given reference dose fractionation is in the range of conventional fractionation.
doi.org/10.1182/blood-2023-191089 , https://doi.org/10.1182/blood-2023-184808
These are only in an abstract form. Full manuscript is not available yet.
doi:10.1016/j.adro.2022.101016
Please see our reference 35, already included in our original manuscript.
doi:10.3390/cancers15102751
This is not a HFRT study.
doi:10.3324/haematol.2023.282804
The median dose/fractionation were 30 Gy (range, 4-50.4 Gy) and 10 fractions (range, 1-28 fractions). Not a HFRT study.
doi:10.3324/haematol.2023.282804
This reference has been added in discussion and highlighted.
ORR is generally considered as CR+PR. Please explain why was SD considered for ORR
Moreover, it’s not clear to me how Deauville 5 could not be considered as PD as the definition of DS5 is “markedly increased uptake or any new lesion (on response evaluation)”
Kindly see the definitions in methods under subheadings outcome measures with the following references ( #16 and 17) .
Barrington, S.F.; Mikhaeel, N.G.; Kostakoglu, L.; Meignan, M.; Hutchings, M.; Müeller, S.P.; Schwartz, L.H.; Zucca, E.; Fisher, R.I.; Trotman, J. Role of imaging in the staging and response assessment of lymphoma: consensus of the International Conference on Malignant Lymphomas Imaging Working Group. Journal of clinical oncology 2014, 32, 3048.
Cheson, B.D.; Fisher, R.I.; Barrington, S.F.; Cavalli, F.; Schwartz, L.H.; Zucca, E.; Lister, T.A. Recommendations for initial evaluation, staging, and response assessment of Hodgkin and non-Hodgkin lymphoma: the Lugano classification. Journal of clinical oncology 2014, 32, 3059.
“of contemporary interventions”: clarify what interventions
This refers to stereotactic ablative radiotherapy (SABR) and hypofractionated RT(HFRT) delivering highly precise and biologically effective radiation doses have been employed in common malignancies, including lung and prostate cancer [7,8] .
Specify is anatomical response and metabolic response were defined according to RECIST and PERCIST criteria.
Kindly see the definitions in methods under subheadings outcome measures with the following references ( #16 and 17) .
Specify the number of patients treated with definitive intent and the type of indolent NHL.
Kindly see table 1. There were 11 patients treated with definitive intent. Definitive intent has been further defined as "Definitive treatment indicates patients exclusively received RT and no chemotherapy.
Specify the timing of chemotherapy,
Chemotherapy details with drugs and mean number of cycles ( 5) has been explained in the results section. Those who received consolidative RT were treated after the completion of chemotherapy.
“Others retrospectively reviewed” specify the name
Name specified and highlighted . ( Ref. 36)Thank you
modern technology” replace with “modern techniques”
Changed and highlighted.
Round 2
Reviewer 3 Report
Comments and Suggestions for Authors
Despite addressing some comments, one crucial question remains unanswered:
1. The rationale behind the selection of hematological malignancies (HM) patients for hypofractionation radiotherapy (HFRT) is not clearly delineated. The authors should provide a comprehensive explanation of the underlying reasons and considerations that led to the application of HFRT in this specific patient population.
Author Response
Dear reviewer
Comment: “The rationale behind the selection of hematological malignancies
(HM) patients for hypofractionation radiotherapy (HFRT) is not clearly
delineated. The authors should provide a comprehensive explanation of
the underlying reasons and considerations that led to the application
of HFRT in this specific patient population."
Response: Many thanks for your comments and feedback. Indeed, this is an important point that forms the basis of this research as explained in the introductory and discussion paragraphs of the manuscript, as below.
In the introduction section:
- There has been a paradigm shift in radiation oncology with rapid advancements in RT techniques, including the introduction of four-dimensional image acquisition, real-time image guidance, and intensity-modulated radiotherapy (IMRT). As such, stereotactic ablative radiotherapy (SABR) and hypofractionated RT(HFRT) delivering highly precise and biologically effective radiation doses have been employed in common malignancies, including lung and prostate cancer [7,8]. The outcome of these contemporary interventions has been impressive, with improved local tumor control, increased overall survival, and minimal toxicity [9]. Further, the reduced number of fractions and an overall shorter treatment time may be cost effective and particularly benefit patients in remote areas away from urban cancer centers [10,11].Therefore, it is intriguing to explore and evaluate the therapeutic efficacy of HFRT in the radiotherapeutic management of HM, assuming no increased toxicity or compromise of local control.
- In March 2020, The COVID-19 pandemic was declared, creating a global healthcare crisis. Human and technical resources were reclassified and redirected to mitigate the challenges of the pandemic. RT departments were forced to rethink and innovate RT delivery models to minimize the risk of COVID-19 infection in cancer patients and healthcare staff. Consequently, several professional groups and organizations suggested alterations to conventional radiation schedules delivered over several weeks. HFRT with a reduced number of fractions, a shorter overall therapy time, and a higher dose per fraction was proposed, with or without clinical evidence, in the radiotherapeutic management of several cancers [12].The intention was to reduce transmission and the risk of infection among immunocompromised cancer patients and involved healthcare workers, and diminish the consequences of reduced human resources during the pandemic [13].
- The International Lymphoma Radiation Oncology Group (ILROG) task force provided emergency guidelines with alternative hypofractionated treatment regimens in response to the COVID-19 pandemic. The proposed dose/fractionation schemes were based on pre-defined radiobiological parameters (i.e., the α/β ratio, total dose in 2 Gy fractions (EQD2), and biological equivalent dose (BED)) to maintain the clinical efficacy and toxicity at levels similar to standard dose fractionation [14].
In the discussion section:
- HM are highly radiosensitive compared to solid malignancies, and have a distinct radiobiological response with early interphase, premitotic, and apoptotic cell death. The possible cellular mechanisms underlying the response to RT leading to apoptosis are lipid peroxidation at the cell membrane, modulation of signal transduction, radiation-induced cross-linking of nuclear DNA, and DNA fragmentation [20].Thus in clinical practice, low to moderate dose involved site radiotherapy (ISRT) with CF; 20-50 Gy over 2-5 weeks) constitutes the standard radiotherapeutic management for most HM with excellent response and long term outcomes.
- The radiosensitivity of lymphomas is remarkable, with a low surviving fraction at 2 Gy (SF2Gy), a high ABR (8-10 Gy), and little to no shoulder on a cell survival curve. [31-34]. During the COVID-19 pandemic, the ILROG designed a set of HFRT dose fractionation schedules to treat HM based on EQD2 with an ABR of 10 and 3 for measurable endpoints of tumor control as well as acute and late toxicity [14].With the provided radiobiological parameters for dose conversion, HFRT is expected to achieve similar tumor control as conventionally fractionated radiotherapy (CF-RT), and any potential late toxicity for organs at risk (OAR) within the irradiation field may be mitigated by the modern conformal RT techniques.
- To date, there is scant literature available on the efficacy of HFRT in HM.
The references contained in the text are available in the submitted revised copy of the manuscript.